# Maximum Likelihood Estimation for Flow Matching by Direct Second-order Trace Objective

## Abstract

Flow matching, one of the attractive deep generative models, has recently been used in wide modality. Despite the remarkable success, the flow matching objective of the vector field is insufficient for maximum likelihood estimation. Previous works show that adding the vector field's high-order gradient objectives further improves likelihood. However, their method only minimizes the upper bound of the high-order objectives, hence it is not guaranteed that the objectives themselves are indeed minimized, resulting in likelihood maximization becoming less effective. In this paper, we propose a method to directly minimize the high-order objective. Since our method guarantees that the objective is indeed minimized, our method is expected to improve likelihood compared to previous works. We verify that our proposed method achieves better likelihood in practice through experiments on 2D synthetic datasets and high-dimensional image datasets.

## 1 Introduction

Diffusion models (Ho et al., 2020; Song et al., 2021b) have shown a high ability to sample from the data distribution in broad modalities, such as image generation (Ho et al., 2020; Song et al., 2021b; Dhariwal & Nichol, 2021; Ho & Salimans, 2021; Rombach et al., 2022), audio generation (Kong et al., 2021; Popov et al., 2021; Chen et al., 2020), and video generation (Ho et al., 2022; Singer et al., 2022). Moreover, to improve the quality of data generation, many previous works have proposed various formulations (Rombach et al., 2022; Karras et al., 2022; Nichol & Dhariwal, 2021; Salimans & Ho, 2022). However, an issue requiring a large number of sampling steps to reduce the discretization error has remained.

Flow matching (Lipman et al., 2023; Liu et al., 2023) is one of the variants of diffusion models, which has achieved the state-of-the-art generated data quality. The main modifications from general diffusion models are parameterization and noise scheduling; the model predicts the vector field, which is the sample's time derivative (velocity), and noises are added to training samples to make its path straight. Although we need the marginal vector field to generate new unseen samples, we can use the vector field conditioned by a data sample in the training objective calculation, which enables us to train flow matching simulation-free as diffusion models. Since the discretization error becomes smaller thanks to the straight-like path, flow matching has been used in broad modalities (Esser et al., 2024; Shi et al., 2023; Le et al., 2023; Vyas et al., 2023).

However, the general flow matching objective is insufficient for maximum likelihood estimation (MLE), or equivalent, minimizing KL divergence between the data distribution and the generated distribution (Zheng et al., 2023). Lu et al. (2022a) prove that the KL divergence is bounded by the integration of the Fisher divergence in ordinary differential equation (ODE) form diffusion models. Moreover, they prove that the Fisher divergence is bounded by an increasing function of errors of high-order gradients of the score function and the model. In flow matching, Zheng et al. (2023) verify the identical theory. However, since the actual values of high-order gradients of the score function or the vector field are intractable, their proposed method only minimizes the upper bound of their objectives, not the objectives themselves. Hence, their methods do not strictly guarantee that the actual objectives necessarily decrease and that the upper bound of the KL divergence is also minimized, resulting in likelihood maximization becoming less effective.

Table 1: The summarization of the differences between our proposed method and previous works. Our proposed method can minimize high-order objectives directly, unlike previous works.

| Method | Model parameterization | Objective function |
|---|---|---|
| Diffusion models | Score function | First-order objective |
| Flow matching | Vector field | First-order objective |
| Lu et al. (2022a) | Score function | Upper bounds of high-order objectives |
| Zheng et al. (2023) | Vector field | Upper bounds of high-order objectives |
| **Ours** | Vector field | High-order objectives **directly** |

In this paper, we propose a method to minimize the second-order objective directly in flow matching. The second-order objective derived in the previous works (Lu et al., 2022a; Zheng et al., 2023) includes the gradient of the marginal score function or the vector field. On the contrary, our method uses the gradient of the conditional one in the second-order objective. Modification from the marginal to the conditional one enables us to calculate the second-order objective simulation-free. Additionally, although the second-order objective is defined by its matrix norm, our proposed method uses its trace instead for efficiency. In contrast to the previous works, our proposed method guarantees that the second-order objective is indeed minimized as long as the training loss decreases since we can calculate the second-order objective itself and minimize it directly. By minimizing the second-order objective itself in all timesteps, we can expect that the value of the function bounding the Fisher divergence is minimized and that the Fisher divergence is also minimized, furthermore, the upper bound of the KL divergence is minimized. To verify the effectiveness of our method in practical settings, we conduct experiments on 2D datasets and high-dimensional image datasets. We show that our method actually improves the likelihood from the original flow matching in 2D synthetic datasets. On image datasets, we show that our method achieves better likelihood than previous works of 3.07 bits/dim of negative log-likelihood on MNIST and 2.62 bits/dim on CIFAR-10, and competitive likelihood of 4.12 bits/dim on ImageNet32×32. Furthermore, we conduct an ablation study to verify that our method minimizing the second-order objective directly indeed maximizes likelihood compared to the methods minimizing its upper bound. Table 1 summarizes the differences between our proposed method and previous works.

## 2 PRELIMINARIES

In this section, we provide preliminaries of diffusion models and flow matching in Sec. 2.1 and 2.2, respectively. Lastly, we provide a unified perspective to connect flow matching to diffusion models in Sec. 2.3.

### 2.1 DIFFUSION MODELS

In diffusion models, the forward diffusion process is defined as the process by which noises are gradually added to a data sample $\mathbf{x}_0$. That process is expressed by the SDE,

$$d\mathbf{x}_t = f(t)\mathbf{x}_t \, dt + g(t) \, d\boldsymbol{w}, \tag{1}$$

where $f : [0, T] \to \mathbb{R}, g : [0, T] \to \mathbb{R}$, and $\boldsymbol{w}$ is the standard Wiener process. Equation (1) has a joint distribution $q_t(\mathbf{x}_t)$ as a solution with an initial value $\mathbf{x}_0 \sim q(\mathbf{x}_0)$. When given an initial value $\mathbf{x}_0$, the conditional distribution $q_t(\mathbf{x}_t|\mathbf{x}_0)$ has a closed form,

$$q_t(\mathbf{x}_t|\mathbf{x}_0) = \mathcal{N}(\mathbf{x}_t; \alpha_t \mathbf{x}_0, \sigma_t^2 \boldsymbol{I}), \tag{2}$$

where $\alpha_t$ and $\sigma_t$ satisfy

$$f(t) = \frac{d \log \alpha_t}{dt}, \quad g(t)^2 = \frac{d\sigma_t^2}{dt} - 2f(t)\sigma_t^2, \tag{3}$$

respectively. The forward process has the following backward process that has the same joint distribution to $q_t(\mathbf{x}_t)$ as a solution,

$$d\mathbf{x}_t = \left( (f(t)\mathbf{x}_t - g(t)^2 \nabla \log q_t(\mathbf{x}_t) \right) \, dt + g(t) \, d\bar{\boldsymbol{w}}, \tag{4}$$

where $\bar{\boldsymbol{w}}$ is the standard Wiener process in reverse time. Although the score function $\nabla \log q_t(\mathbf{x}_t)$ is generally intractable, a model $\boldsymbol{s}(\mathbf{x}_t, t)$ with parameter $\boldsymbol{\theta}$ can approximate $\nabla \log q_t(\mathbf{x}_t)$ via

$$\underset{\boldsymbol{\theta}}{\text{minimize}} \quad \mathbb{E}_{\mathbf{x}_0 \sim q(\mathbf{x}_0), \boldsymbol{\epsilon} \sim \mathcal{N}(\mathbf{0}, \boldsymbol{I})} \left[ \| \boldsymbol{s}_{\boldsymbol{\theta}}(\alpha_t \mathbf{x}_0 + \sigma_t \boldsymbol{\epsilon}, t) - \nabla \log q_t(\mathbf{x}_t | \mathbf{x}_0) \|_2^2 \right] \tag{5}$$

$$= \underset{\boldsymbol{\theta}}{\text{minimize}} \quad \mathbb{E}_{\mathbf{x}_0 \sim q(\mathbf{x}_0), \boldsymbol{\epsilon} \sim \mathcal{N}(\mathbf{0}, \boldsymbol{I})} \left[ \left\| \boldsymbol{s}_{\boldsymbol{\theta}}(\alpha_t \mathbf{x}_0 + \sigma_t \boldsymbol{\epsilon}, t) + \frac{1}{\sigma_t} \boldsymbol{\epsilon} \right\|_2^2 \right]. \tag{6}$$

Moreover, there exists an ODE (7) that has the same joint distribution $q_t(\mathbf{x}_t)$ to the SDE (4) with initial values $\mathbf{x}_T \sim \mathcal{N}(\mathbf{x}_T; \mathbf{0}, \boldsymbol{I})$,

$$d\mathbf{x}_t = \left[ f(t)\mathbf{x}_t - \frac{1}{2} g(t)^2 \nabla \log q_t(\mathbf{x}_t) \right] dt. \tag{7}$$

In the inference phase, we can obtain generated samples by solving SDE (4) or ODE (7) with any solvers (Karras et al., 2022; Song et al., 2021a; Lu et al., 2022b;c). Denoting $p_t(\mathbf{x}_t)$ as the distribution of $\mathbf{x}_t$ obtained by solving SDE (4) or ODE (7), it is expected $p_t(\mathbf{x}_t) \approx q_t(\mathbf{x}_t)$ holds, and further $p_0(\mathbf{x}_0) \approx q_0(\mathbf{x}_0) = q(\mathbf{x}_0)$ does.

## 2.2 FLOW MATCHING

Flow matching (Lipman et al., 2023; Liu et al., 2023) was proposed as a variant of continuous normalizing flows(CNFs) or neural ODE (Chen et al., 2018; Liu et al., 2023). We derive the flow matching method following the derivation in Lipman et al. (2023).

We denote $\mathbf{x}_0 \in \mathbb{R}^n$ and $q(\mathbf{x}_0)$ as a data sample [1] and the true data distribution, respectively. We first formulate a time-dependant distribution $q_t(\mathbf{x}_t)$ for $t \in [0, T]$ using the conditional distribution $q_t(\mathbf{x}_t | \mathbf{x}_0)$ as

$$q_t(\mathbf{x}_t) = \int q_t(\mathbf{x}_t | \mathbf{x}_0) q(\mathbf{x}_0) d\mathbf{x}_0, \tag{8}$$

$$q_t(\mathbf{x}_t | \mathbf{x}_0) = \mathcal{N} \left( \mathbf{x}_t; \alpha_t \mathbf{x}_0, \sigma_t^2 \boldsymbol{I} \right), \tag{9}$$

$$\alpha_t = 1 - \frac{t}{T}, \quad \sigma_t = \frac{t + (T - t)\sigma_{\min}}{T}. \tag{10}$$

where $\sigma_{\min} > 0$ is a small value to avoid numerical issues. At $t \in (0, T)$, we can sample $\mathbf{x}_t | \mathbf{x}_0 \sim q_t(\mathbf{x}_t | \mathbf{x}_0)$ via reparameterization trick as,

$$\mathbf{x}_t = \alpha_t \mathbf{x}_0 + \sigma_t \boldsymbol{\epsilon}, \tag{11}$$

where $\boldsymbol{\epsilon} \sim \mathcal{N}(\mathbf{x}; \mathbf{0}, \boldsymbol{I})$. In flow matching, we generate samples from $q_t(\mathbf{x}_t)$ by following a vector field starting from noises. We first define the vector field $\boldsymbol{u}_t(\mathbf{x}_t)$ as the time derivative of the sample $\mathbf{x}_t$

$$\boldsymbol{u}_t(\mathbf{x}_t) := \frac{d\mathbf{x}_t}{dt}. \tag{12}$$

However, we cannot calculate $\boldsymbol{u}_t(\mathbf{x}_t)$ since we cannot sample $\mathbf{x}_t$ from the marginal distribution $q_t(\mathbf{x}_t)$.

Therefore, we consider to use a neural network $\boldsymbol{v}_{\boldsymbol{\theta}}(\mathbf{x}_t, t)$ with parameter $\boldsymbol{\theta}$ instead of $\boldsymbol{u}_t(\mathbf{x}_t)$. As a preparation to approximate $\boldsymbol{u}_t(\mathbf{x}_t)$ by $\boldsymbol{v}_{\boldsymbol{\theta}}(\mathbf{x}_t, t)$, we define the conditional vector field $\boldsymbol{u}_t(\mathbf{x}_t | \mathbf{x}_0)$ as

$$\boldsymbol{u}_t(\mathbf{x}_t | \mathbf{x}_0) := \left. \frac{d\mathbf{x}_t | \mathbf{x}_0}{dt} \right|_{\mathbf{x}_0} = \dot{\alpha}_t \mathbf{x}_0 + \dot{\sigma}_t \frac{\mathbf{x}_t - \alpha_t \mathbf{x}_0}{\sigma_t} = (1 - \sigma_{\min}) \boldsymbol{\epsilon} - \mathbf{x}_0, \tag{13}$$

where $\dot{\alpha}_t$ and $\dot{\sigma}_t$ mean the time derivative of $\alpha_t$ and $\sigma_t$, respectively. Then, we can learn $\boldsymbol{u}(\mathbf{x}_t)$ by $\boldsymbol{v}_{\boldsymbol{\theta}}(\mathbf{x}_t, t)$ through

$$\underset{\boldsymbol{\theta}}{\text{minimize}} \quad \mathbb{E}_{\mathbf{x}_0 \sim q(\mathbf{x}_0), \boldsymbol{\epsilon} \sim \mathcal{N}(\mathbf{x}; \mathbf{0}, \boldsymbol{I}), \mathbf{x}_t \sim q_t(\mathbf{x}_t | \mathbf{x}_0)} \left[ \| \boldsymbol{v}_{\boldsymbol{\theta}}(\mathbf{x}_t, t) - \boldsymbol{u}_t(\mathbf{x}_t | \mathbf{x}_0) \|_2^2 \right]. \tag{14}$$

---

[1]Note that in Lipman et al. (2023) set the range of $t$ to $[0, 1]$ and let $\mathbf{x}_1$ be a clean sample, but we changed the range to $[0, T]$ and $\mathbf{x}_0$ be a clean sample to align to the notation of diffusion models.

since the following holds

$$\nabla_{\boldsymbol{\theta}} \mathbb{E}_{\mathbf{x}_t \sim q_t(\mathbf{x}_t)} \left[ \| \boldsymbol{v}_{\boldsymbol{\theta}}(\mathbf{x}_t, t) - \boldsymbol{u}_t(\mathbf{x}_t) \|_2^2 \right]$$

$$= \nabla_{\boldsymbol{\theta}} \mathbb{E}_{\mathbf{x}_0 \sim q(\mathbf{x}_0), \boldsymbol{\epsilon} \sim \mathcal{N}(\mathbf{x}; \mathbf{0}, \boldsymbol{I}), \mathbf{x}_t \sim q_t(\mathbf{x}_t | \mathbf{x}_0)} \left[ \| \boldsymbol{v}_{\boldsymbol{\theta}}(\mathbf{x}_t, t) - \boldsymbol{u}_t(\mathbf{x}_t | \mathbf{x}_0) \|_2^2 \right]. \tag{15}$$

In the inference phase, we can obtain the generated sample by solving the differential equation

$$\frac{d\mathbf{x}_t}{dt} = \boldsymbol{v}_{\boldsymbol{\theta}}(\mathbf{x}_t, t) \tag{16}$$

by an arbitrary solver such as the Euler method with an initial value $\mathbf{x}_0 \sim \mathcal{N}(\mathbf{0}, \boldsymbol{I})$. Similarly to diffusion models, denoting $p_t(\mathbf{x}_t)$ as the distribution of $\mathbf{x}_t$ by solving ODE (16), it is expected $p_1(\mathbf{x}_0) \approx q_1(\mathbf{x}_0) \approx q(\mathbf{x}_0)$ holds.

### 2.3 CONNECTION TO DIFFUSION MODELS

Now, we can interpret flow matching as a variant of diffusion models with two modifications. The first modification is the model parameterization. In diffusion models, the model predicts the score function $\nabla \log q_t(\mathbf{x}_t)$. On the other hand, in flow matching, the model predicts the vector field $\boldsymbol{u}_t(\mathbf{x}_t)$, and $\boldsymbol{u}_t(\mathbf{x}_t)$ is formulated by Equations (7) and (12) as

$$\boldsymbol{u}_t(\mathbf{x}_t) = f(t)\mathbf{x}_t - \frac{1}{2}g(t)^2 \nabla \log q_t(\mathbf{x}_t), \quad \boldsymbol{v}_{\boldsymbol{\theta}}(\mathbf{x}_t, t) = f(t)\mathbf{x}_t - \frac{1}{2}g(t)^2 \boldsymbol{s}_{\boldsymbol{\theta}}(\mathbf{x}_t, t). \tag{17}$$

The second modification is the scheduling of the ratio of noises in Equation (2). While $\alpha_t$ and $\sigma_t$ are defined as non-linear functions in diffusion models (Ho et al., 2020; Karras et al., 2022), they are defined as linear functions of $t$ by Equation (10) in flow matching. Therefore, we can perceive flow matching as the ODE form diffusion models.

## 3 METHOD

We first show how to maximize likelihood in diffusion models with ODE form in Sec. 3.1, and that MLE with ODE form has an additional term to the SDE form, following Lu et al. (2022a). Subsequently, we explain Lu et al. (2022a) method, which minimizes the additional term by minimizing upper bounds of high-order objectives in Sec. 3.2. Lastly, we present our method to minimize the second-order objective directly.

### 3.1 MLE FOR DIFFUSION MODELS

MLE is identical to minimizing KL divergence between the data distribution and the generated distribution. Song et al. (2021b) show that denoising score matching minimizes the KL divergence by maximizing the evidence lower bound (ELBO) of likelihood, which is identical to Equation (6). However, Lu et al. (2022a) show that ODE form diffusion models have a different ELBO.

Specifically, they show that KL divergence between the data distribution $q_0$ and the generated distribution $p_0$ can be bounded as

$$D_{\mathrm{KL}}(q_0 \| p_0) = D_{\mathrm{KL}}(q_T \| p_T) + \mathcal{J}_{\mathrm{ODE}} \tag{18}$$

$$\leq D_{\mathrm{KL}}(q_T \| p_T) + \sqrt{\mathcal{J}_{\mathrm{SM}}} \cdot \sqrt{\mathcal{J}_{\mathrm{Fisher}}}, \tag{19}$$

where

$$\mathcal{J}_{\mathrm{ODE}} = \frac{1}{2} \int_0^T g(t)^2 \mathbb{E}_{\mathbf{x}_t} \left[ (\boldsymbol{s}_{\boldsymbol{\theta}}(\mathbf{x}_t, t) - \nabla \log q_t(\mathbf{x}_t))^T (\nabla \log p_t(\mathbf{x}_t) - \nabla \log q_t(\mathbf{x}_t)) \right] dt, \tag{20}$$

$$\mathcal{J}_{\mathrm{SM}} = \frac{1}{2} \int_0^T g(t)^2 \mathbb{E}_{\mathbf{x}_t} \left[ \| \boldsymbol{s}_{\boldsymbol{\theta}}(\mathbf{x}_t, t) - \nabla \log q_t(\mathbf{x}_t) \|_2^2 \right] dt, \tag{21}$$

$$\mathcal{J}_{\mathrm{Fisher}} = \frac{1}{2} \int_0^T g(t)^2 \mathbb{E}_{\mathbf{x}_t} \left[ \| \nabla \log p_t(\mathbf{x}_t) - \nabla \log q_t(\mathbf{x}_t) \|_2^2 \right] dt. \tag{22}$$

Hence, the general objective function (6) only minimizes $\mathcal{J}_{\mathrm{SM}}$, however, we can further minimize the KL divergence by additionally minimizing $\mathcal{J}_{\mathrm{Fisher}}$.

## 3.2 HIGH-ORDER DIFFUSION MODELS

Lu et al. (2022a) propose to minimize high-order objectives to minimize $\mathcal{J}_{\text{Fisher}}$. Strictly, they show that the Fisher divergence $\mathbb{E}_{\mathbf{x}_t}[\|\nabla \log p_t(\mathbf{x}_t) - \nabla \log q_t(\mathbf{x}_t)\|_2^2]$, which is the integrand of $\mathcal{J}_{\text{Fisher}}$, is bounded by a function $U(t; \delta_1, \delta_2, \delta_3, q)$ with some assumptions, where

$$\|\boldsymbol{s}_{\boldsymbol{\theta}}(\mathbf{x}_t, t) - \nabla \log q_t(\mathbf{x}_t)\|_2 \le \delta_1, \tag{23}$$

$$\|\nabla \boldsymbol{s}_{\boldsymbol{\theta}}(\mathbf{x}_t, t) - \nabla^2 \log q_t(\mathbf{x}_t)\|_F \le \delta_2, \tag{24}$$

$$\|\nabla \operatorname{div} \boldsymbol{s}_{\boldsymbol{\theta}}(\mathbf{x}_t, t) - \nabla \operatorname{div} \log q_t(\mathbf{x}_t)\|_2 \le \delta_3. \tag{25}$$

Since $U(t; \delta_1, \delta_2, \delta_3, q)$ is a strictly increasing function of $\delta_1, \delta_2$, and $\delta_3$, we can minimize $\mathcal{J}_{\text{Fisher}}$ by minimizing the left-hand sides of Equations (23), (24), and (25). However, since $\nabla \log q_t(\mathbf{x}_t)$ and its high-order gradients are intractable, they propose to minimize their upper bounds in practice by minimizing the following objectives;

$$\mathbb{E}_{\mathbf{x}_0, \mathbf{x}_t} \left[ \|\boldsymbol{s}_{\boldsymbol{\theta}}(\mathbf{x}_t, t) - \nabla \log q_t(\mathbf{x}_t|\mathbf{x}_0)\|_2^2 \right], \tag{26}$$

$$\mathbb{E}_{\mathbf{x}_0, \mathbf{x}_t} \left[ \left\|\nabla \boldsymbol{s}_{\boldsymbol{\theta}}(\mathbf{x}_t, t) - \nabla^2 \log q_t(\mathbf{x}_t|\mathbf{x}_0) - \boldsymbol{\ell}_1 \boldsymbol{\ell}_1^T \right\|_F^2 \right], \tag{27}$$

$$\mathbb{E}_{\mathbf{x}_0, \mathbf{x}_t} \left[ \left| \operatorname{div} \boldsymbol{s}_{\boldsymbol{\theta}}(\mathbf{x}_t, t) - \operatorname{div} \nabla \log q_t(\mathbf{x}_t|\mathbf{x}_0) - \|\boldsymbol{\ell}_1\|_2^2 \right|^2 \right], \tag{28}$$

$$\mathbb{E}_{\mathbf{x}_0, \mathbf{x}_t} \left[ \|\nabla \operatorname{div} \boldsymbol{s}_{\boldsymbol{\theta}}(\mathbf{x}_t, t) - \boldsymbol{\ell}_3\|^2 \right], \tag{29}$$

where

$$\boldsymbol{\ell}_1 = \boldsymbol{s}_{\boldsymbol{\theta}}^{sg}(\mathbf{x}_t, t) - \nabla \log q_t(\mathbf{x}_t|\mathbf{x}_0), \tag{30}$$

$$\boldsymbol{\ell}_2 = \nabla \boldsymbol{s}_{\boldsymbol{\theta}}^{sg}(\mathbf{x}_t, t) - \nabla^2 \log q_t(\mathbf{x}_t|\mathbf{x}_0), \tag{31}$$

$$\boldsymbol{\ell}_3 = \left( \|\boldsymbol{\ell}_1\|_2^2 \boldsymbol{I} - \operatorname{tr}(\boldsymbol{\ell}_2)\boldsymbol{I} - 2\boldsymbol{\ell}_2 \right) \boldsymbol{\ell}_1 \tag{32}$$

and $sg$ is the stop gradient operator.

## 3.3 DIRECT HIGH-ORDER FLOW MATCHING

The general flow matching objective (14) is not derived for the purpose of MLE. As flow matching is a variant of the ODE form diffusion models as described in Sec. 2.3, we can apply the previous work described in Sec. 3.2 to flow matching. By considering the description in Sec. 3.1 in flow matching formulation, although the general flow matching objective (14) can minimize the KL divergence, we can further minimize it by minimizing $\mathcal{J}_{\text{Fisher}}$. As described in 3.2, we need to minimize high-order objectives to minimize $\mathcal{J}_{\text{Fisher}}$. Zheng et al. (2023) already propose to minimize high-order objectives in flow matching with several techniques.

However, Lu et al. (2022a) and Zheng et al. (2023) proposed methods minimize the upper bounds of the high-order objectives, not objectives themselves. Hence, their methods do not necessarily minimize the upper bound of the KL divergence, as a result, their methods reduce the effect of likelihood maximization. Then, we propose a method to minimize the high-order flow matching objectives directly.

First, we show that the Fisher divergence is bounded by a function of high-order flow matching objectives, similarly to Sec. 3.2.

**Theorem 3.1** (Proof in Appendix A). *Let $q_t(\mathbf{x}_t), p_t(\mathbf{x}_t)$ be a joint distribution generated by the true vector field $\boldsymbol{u}_t(\mathbf{x}_t)$ and a model $\boldsymbol{v}_{\boldsymbol{\theta}}(\mathbf{x}_t, t)$, respectively. Assume that there exists $C \in \mathbb{R}$ such that $\|\nabla^2 \log p_t(\mathbf{x}_t)\|_2 < C$. Then, there exists a function $U(t, \delta_1, \delta_2, \delta_3, C, q_t)$ which is an strictly increasing function of $\delta_1, \delta_2$ and $\delta_3$ such that*

$$\|\nabla \log p_t(\mathbf{x}_t) - \nabla \log q_t(\mathbf{x}_t)\|_2^2 \le U(t, \delta_1, \delta_2, \delta_3, C, q_t)$$

*where $\delta_1, \delta_2$ and $\delta_3$ satisfy*

$$\|\boldsymbol{v}_{\boldsymbol{\theta}}(\mathbf{x}_t, t) - \boldsymbol{u}(\mathbf{x}_t)\|_2 \le \delta_1,$$
$$\|\nabla \boldsymbol{v}_{\boldsymbol{\theta}}(\mathbf{x}_t, t) - \nabla \boldsymbol{u}(\mathbf{x}_t|\mathbf{x}_0)\|_F \le \delta_2,$$
$$\|\nabla \operatorname{div} \boldsymbol{v}_{\boldsymbol{\theta}}(\mathbf{x}_t, t) - \nabla \operatorname{div} \boldsymbol{u}(\mathbf{x}_t)\|_2 \le \delta_3.$$

The notable point of Theorem 3.1 is using $\nabla \boldsymbol{u}_t(\mathbf{x}_t|\mathbf{x}_0)$ instead of $\nabla \boldsymbol{u}_t(\mathbf{x}_t)$ in the second-order objective. As described in Sec. 3.2, the previous works minimize its upper bound since $\nabla \boldsymbol{u}_t(\mathbf{x}_t)$ is intractable. On the contrary, Theorem 3.2 enables us to minimize the second-order objective directly. By directly minimizing the second-order objective $\delta_2$ directly, we can guarantee that $U$ is indeed minimized while the previous works cannot. Furthermore, we can expect that the Fisher divergence is minimized, and that the upper bound of the KL divergence is also minimized more tightly than the previous works.

Then, we add the second-order objective to the original flow matching objective (14). For the efficiency, we minimize the trace of the second-order objective instead of the matrix norm. We do not minimize the third-order objective since the we want computational time not to be longer and the contribution of directly minimizing the second-order objective is more significant. In summary, the whole objective of our proposed method is

$$\mathbb{E}_{\mathbf{x}_0, \boldsymbol{\epsilon}, \mathbf{x}_t} \left[ \|\boldsymbol{v}_{\boldsymbol{\theta}}(\mathbf{x}_t, t) - \boldsymbol{u}_t(\mathbf{x}_t|\mathbf{x}_0)\|_2^2 + \lambda_{\mathrm{div}} |\mathrm{div}\, \boldsymbol{v}_{\boldsymbol{\theta}}(\mathbf{x}_t, t) - \mathrm{div}\, \boldsymbol{u}_t(\mathbf{x}_t|\mathbf{x}_0)|^2 \right] \tag{33}$$

where $\lambda_{\mathrm{div}} \in \mathbb{R}$ is a hyper-parameter which stands for a weight of second-order objective.

To compute $\mathrm{div}\, \boldsymbol{v}_{\boldsymbol{\theta}}(\mathbf{x}_t, t)$, we use Hutchinson's trace estimation method (Hutchinson, 1989). Hutchinson's trace estimation method approximates the trace of a matrix $\boldsymbol{A}$ by the Monte Carlo method,

$$\mathrm{tr}\, \boldsymbol{A} = \mathbb{E}_{\boldsymbol{w} \sim p(\boldsymbol{w})}[\boldsymbol{w}^T \boldsymbol{A} \boldsymbol{w}] \approx \frac{1}{n} \sum_{i=1}^{n} \boldsymbol{w}_i^T \boldsymbol{A} \boldsymbol{w}_i, \tag{34}$$

where $n$ is a number of sampling $\boldsymbol{w}$, and $p(\boldsymbol{w})$ is a multivariate standard normal distribution or multivariate Rademacher distribution, whose element takes $-1$ or $1$ uniformly. By using the Hutchinson's method, we can approximate $\mathrm{div}\, \boldsymbol{v}_{\boldsymbol{\theta}}(\mathbf{x}_t, t)$ as

$$\mathrm{div}\, \boldsymbol{v}_{\boldsymbol{\theta}}(\mathbf{x}_t, t) \approx \frac{1}{n} \sum_{i=1}^{n} \boldsymbol{w}_i^T \nabla(\boldsymbol{v}_{\boldsymbol{\theta}}(\mathbf{x}_t, t)) \boldsymbol{w}_i = \frac{1}{n} \sum_{i=1}^{n} \nabla(\boldsymbol{w}_i^T \boldsymbol{v}_{\boldsymbol{\theta}}(\mathbf{x}_t, t)) \boldsymbol{w}_i. \tag{35}$$

Since $\boldsymbol{w}_i^T \boldsymbol{v}_{\boldsymbol{\theta}}(\mathbf{x}_t, t)$ is a scalar, we can calculate $\nabla(\boldsymbol{w}_i^T \boldsymbol{v}_{\boldsymbol{\theta}}(\mathbf{x}_t, t))$ precisely as a vector-Jacobian product in the automatic differentiation framework. We adopted $n = 1$ to avoid the high computational costs.

# 4 RELATED WORK

**Flow matching.** One of the advantages of flow matching is the straight-like path, which mitigates the discretization error in the inference phase. Thanks to that feature, flow matching achieved state-of-the-art quality in image generation (Lipman et al., 2023; Liu et al., 2023). Sequentially, flow matching has been used in broad modalities, such as image generation (Esser et al., 2024; Dao et al., 2023; Yan et al., 2024), audio generation (Mehta et al., 2024; Liu et al., 2024; Prajwal et al., 2024), and discrete data generation (Gat et al., 2024; Nisonoff et al., 2024). Regarding the training theory of flow matching, there are some works regarding theoretical error bounds in terms of Wasserstein distance (Benton et al., 2023; Fukumizu et al., 2024). To minimize the KL divergence in flow matching, although Zheng et al. (2023) proposed some techniques, including minimizing second-order objective, they only minimize an upper bound of the objective instead of the objective itself, as described in Sec. 3.3.

**MLE for diffusion models.** Song et al. (2021b) showed that minimizing the denoising score matching loss (6) corresponded to maximizing the ELBO of likelihood. Unlike the theory, it was known that DDPM (Ho et al., 2020), which minimized the ELBO weighted by $\sigma_t^2$, achieved better results than denoising score matching experimentally. Kingma & Gao (2024) showed that the DDPM objective can be perceived as the ELBO with data augmentation of noise adding, and unified previous works proposing other objectives (Nichol & Dhariwal, 2021; Karras et al., 2022; Salimans & Ho, 2022). However, Lu et al. (2022a) verified that ODE form diffusion models have different ELBO, and that we can further minimize the KL divergence by minimizing high-order objectives. However, as described in Sec. 3.3, their method also minimizes the upper bounds of the objectives, not the objectives themselves.

Table 2: The comparison of negative log-likelihood (NLL) with three datasets by the original flow matching and our proposed method. The lower is better. Our proposed method improves the likelihood compared to the original flow matching.

| Method | 8gaussians | moons | 2circles |
|---|---|---|---|
| Flow matching (Lipman et al., 2023) | 3.80 | 2.64 | 3.12 |
| **Ours** | **3.77** | **2.61** | **2.67** |

Table 3: The comparison of 2-Wasserstein distance with three datasets by the original flow matching and our proposed method. The lower is better. Our proposed method outperforms the original flow matching on two datasets and is comparable on another dataset.

| Method | 8gaussians | moons | 2circles |
|---|---|---|---|
| Flow matching (Lipman et al., 2023) | 0.27 | **0.10** | 0.097 |
| **Ours** | **0.20** | 0.12 | **0.063** |

## 5 EXPERIMENTS

In this section, we provide experimental verification of our proposed method through experiments. In Sec. 5.1, we conduct experiments on 2D synthetic datasets as simple toy datasets. In Sec. 5.2, we conduct experiments on image datasets, MNIST, CIFAR-10, and ImageNet32, as high-dimensional datasets.

### 5.1 EXPERIMENTS ON 2D SYNTHETIC DATASET

**Experimental Setup.** We prepare three 2D synthetic datasets, *8gaussians*, *moons*, and *2circles* from two libraries, `torchcfm` (Tong et al., 2024) and `scikit-learn` (Pedregosa et al., 2011). They have 2-dimensional data following each defined probabilistic distribution. In the inference phase, we use the Euler method with 100 steps to generate samples. We use two evaluation measures, negative log-likelihood (NLL) and 2-Wasserstein distance with 1000 samples. We calculate NLL following the technique in Lipman et al. (2023) (please visit Appendix C in Lipman et al. (2023) for more detail). All experiments were conducted on a single V100 GPU. More details of the training setting are provided in Appendix B.1.

**Results.** Table 2 shows the comparison of NLL by original flow matching and our proposed method in three datasets. Our proposed method has better NLL than the original flow matching in all three datasets. Table 3 shows the comparison of 2-Wasserstein distances between the training data and generated data by each method. Our proposed method has smaller distances than the original flow matching in *8gaussians* and *2circles* datasets. While our proposed method has the larger distance in *moons* dataset, the distance is competitive to the original flow matching.

Figure 1 shows 1,000 generated samples by original flow matching (second row) and our proposed method (third row) for each dataset. The first row shows samples from each training dataset. The more yellow color indicates that samples are denser. Our proposed method generates fewer samples in the areas where training samples do not exist (red box) than the original flow matching. That is, our proposed method is more likely not to generate samples far away from the training samples, and vice versa, to generate samples closer to the training samples. This observation means that likelihood of our proposed method is higher than the original flow matching. We can justify qualitatively that our proposed method maximizes likelihood from these results.

### 5.2 EXPERIMENTS ON IMAGE DATASETS

**Experimental Setup.** We prepare three image dataset, MNIST (Deng, 2012), CIFAR-10 (Krizhevsky et al., 2009), and ImageNet32×32 (Chrabaszcz et al., 2017). Each dataset has 60,000, 50,000, and 14,197,122 training images, respectively. We prepared ImageNet32×32 by resizing the images in ImageNet (Deng et al., 2009) to the size of 32×32 following Chrabaszcz et al.

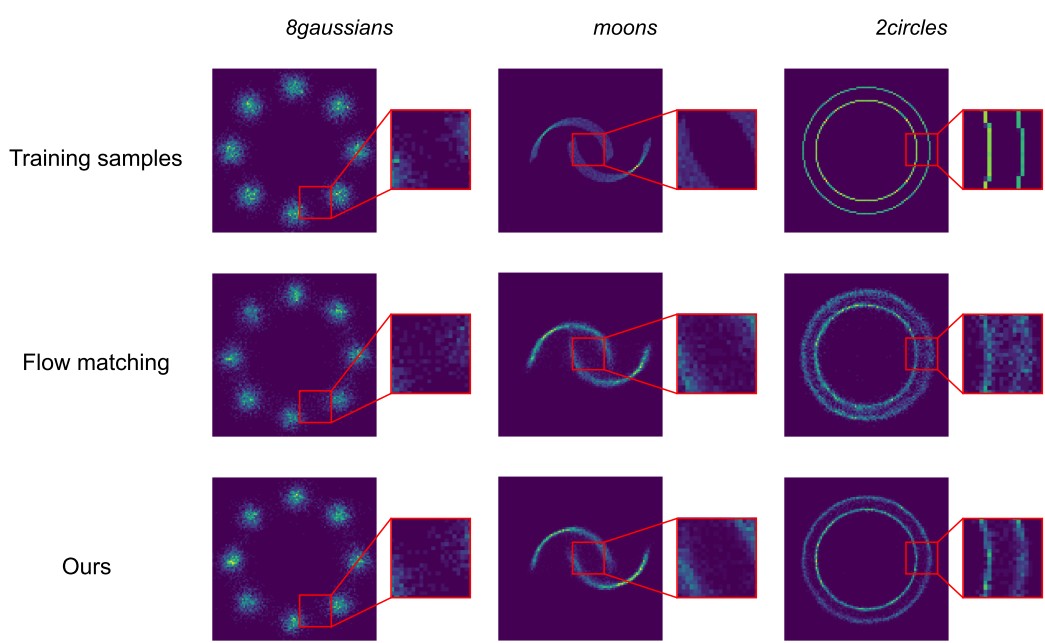

Figure 1: Generated samples by the original flow matching and our proposed method. Our proposed method generates fewer samples in the areas where training samples do not exist (red box) compared to the original flow matching, which implies that our proposed method has a better likelihood than the original flow matching.

Table 4: The comparison of NLL with three datasets by previous works and our proposed method. The lower is better. Our proposed method is comparable to previous works on all datasets.

| Method | MNIST | CIFAR-10 | ImageNet |
|---|---|---|---|
| DDPM (Ho et al., 2020) | - | $\leq 3.75$ | - |
| Score matching (Song et al., 2021b) | - | 3.45 | 4.21 |
| High-order score matching (deep, second) (Lu et al., 2022a) | - | 3.35 | 4.05 |
| High-order score matching (deep, third) | - | 3.27 | 4.03 |
| i-DODE (SP) (Zheng et al., 2023) | - | 2.56 | 3.44 |
| i-DODE (VP) | - | 2.57 | 3.43 |
| i-DODE (VP, with data augmentation) | - | **2.42** | - |
| Flow matching (Lipman et al., 2023) | - | 2.99 | 3.53 |
| Flow matching (reproduced) | 3.10 | 2.68 | 4.07 |
| **Ours** | **3.07** | 2.62 | 4.12 |

(2017). We evaluate our proposed method by NLL. All experiments were conducted on eight A100 GPUs. More details of the training setting are provided in Appendix B.2.

**Results.** Table 4 shows the comparison of NLL with three datasets by previous works and our proposed method. Although our proposed method does not improve NLL from the original flow matching on ImageNet32×32, it improves NLL on MNIST and CIFAR-10. On CIFAR-10, although the best NLL is reported by i-DODE (Zheng et al., 2023), NLL by our proposed method is better than that by the original flow matching. This gap comes from other improvement techniques of i-DODE, as we discuss in Sec. 5.2.1. Through these quantitative results, we show that our proposed method improves likelihood from the original flow matching. We provide qualitative results with the original flow matching and our proposed method in Appendix C.

Table 5: The comparison of NLL by four methods. The lower is better. Our proposed method outperforms the original flow matching and the methods minimizing the upper bound of the second-order objectives.

| Method | MNIST | CIFAR-10 |
|---|---|---|
| Flow matching (reproduced) | 3.10 | 2.68 |
| Upper bound by the matrix norm | 3.12 | 3.08 |
| Upper bound by the trace | 3.09 | 2.77 |
| **Ours** | **3.07** | **2.62** |

### 5.2.1 ABLATION STUDY

In this section, we investigate how directly minimizing the objectives is effective compared to minimizing the upper bound. As described in Sec. 3, we need to minimize the high-order objectives to minimize the upper bound of the KL divergence. Furthermore, our proposed method minimizes the second-order objective directly, while the previous works (Lu et al., 2022a; Zheng et al., 2023) minimize its upper bound. We verify that directly minimizing the objective improves likelihood than minimizing its upper bound through an experiment on image datasets, MNIST and CIFAR-10.

We compare four methods: original flow matching, MLE by minimizing the upper bound of the second-order matrix norm objective, MLE by minimizing the upper bound of the second-order trace objective, and MLE by minimizing the second-order trace objective directly (ours). To calculate the matrix norm, we used an equality of $\|A\|_F^2 = \text{tr}(A^T A)$ and Hutchinson's method, following Lu et al. (2022a). We investigated various values of weight parameter $\lambda_{\text{div}}$ for each method and recorded the best NLL.

Table 5 shows the comparison of NLL by four methods. The methods minimizing the upper bound have worse likelihoods on either dataset than the original flow matching. On the contrary, our proposed method achieves the best NLL thanks to minimizing the second-order trace objective directly. Therefore, we can justify that directly minimizing the second-order objective maximizes likelihood more than minimizing the upper bound. We additionally emphasize that i-DODE(SP), which has better NLL than our proposed method on CIFAR-10 as shown in Table 4, includes minimizing the upper bound of the second-order trace objective. From this result, our method may further improve NLL by combining techniques proposed in previous works with our proposed method. Verification of this hypothesis is the subject of future work.

## 6 DISCUSSIONS AND CONCLUSIONS

Our proposed method has several weak points. First, our proposed method makes training time longer since additional backpropagation process is required in our proposed method. For instance, while the training time of original flow matching is about 16 A100 hours, our proposed method is about 20 A100 hours, which is 1.25 times longer on CIFAR-10 and 2 times longer on ImageNet32. Additionally, usage memory size also increases. However, since the training time increases linearly with the dimension of the model's hidden layers, it is realistically possible to scale our method for larger models. Second, the improvement of NLL by our proposed method is not large. Since the improvement in likelihood does not mean an improvement in image quality, it is unclear whether our proposed method is useful in practical applications. Lastly, our proposed method has general weak points of flow matching, e.g., requiring iterative calculations in the inference phase.

We proposed a method to directly minimize the second-order flow matching objective in addition to the original flow matching objective. Our proposed method guarantees that it minimizes the upper bound of KL divergence between the data distribution and the generated distribution by directly minimizing the second-order flow matching objective, while previous works do not since they minimize only the upper bound of the second-order objective. We verified that our proposed method achieves competitive likelihood with previous works through experiments with 2D synthetic datasets and image datasets. Moreover, we showed that directly minimizing the second-order objective indeed improves the likelihood more than minimizing its upper bound through the ablation study. Furthermore, we showed that our proposed method can be potentially improved by combining techniques

proposed in previous works. We expect our work will further enrich the learning theory of flow matching.

## REPRODUCIBILITY STATEMENT

We provide supplemental information for reproducibility in appendices. In Appendix A, we provide complete proof of theorems in the derivation of our method. In Appendix B, we provide more detailed settings for our implementation in the experiments on image datasets. In Appendix C, we provide qualitative comparisons of our proposed method with the original flow matching.

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

## A  PROOF OF THEOREM 3.1

In this section, we provide proof of Theorem 3.1. Since our theorems are based on Theorem 3.1 and Theorem 3.2 in Lu et al. (2022a), we basically rewrite their proof as flow matching.

First, we derive the upper bound of the KL divergence between the data distribution $q_1$ and the generated distribution $p_1$.

**Lemma A.1.**

$$D_{\mathrm{KL}}(q_0\|p_0) = D_{\mathrm{KL}}(q_T\|p_T) + \mathcal{J}_{\mathrm{ODE}}$$

$$\leq D_{\mathrm{KL}}(q_T\|p_T) + \sqrt{\mathcal{J}_{\mathrm{FM}}} \cdot \sqrt{\mathcal{J}_{\mathrm{Fisher}}},$$

*where*

$$\mathcal{J}_{\mathrm{ODE}} = \int_0^T \mathbb{E}_{\mathbf{x}_t} \left[ (\boldsymbol{v}_{\boldsymbol{\theta}}(\mathbf{x}_t, t) - \boldsymbol{u}_t(\mathbf{x}_t))^T (\nabla \log p_t(\mathbf{x}_t) - \nabla \log q_t(\mathbf{x}_t)) \right] dt,$$

$$\mathcal{J}_{\mathrm{FM}} = \int_0^T \mathbb{E}_{\mathbf{x}_t} \left[ \|\boldsymbol{v}_{\boldsymbol{\theta}}(\mathbf{x}_t, t) - \boldsymbol{u}_t(\mathbf{x}_t)\|_2^2 \right] dt,$$

$$\mathcal{J}_{\mathrm{Fisher}} = \int_0^T \mathbb{E}_{\mathbf{x}_t} \left[ \|\nabla \log p_t(\mathbf{x}_t) - \nabla \log q_t(\mathbf{x}_t)\|_2^2 \right] dt.$$

*Proof.* First, we express $D_{\mathrm{KL}}(q_1\|p_1)$ by integral form as

$$D_{\mathrm{KL}}(q_0\|p_0) = D_{\mathrm{KL}}(q_T\|p_T) + D_{\mathrm{KL}}(q_0\|p_0) - D_{\mathrm{KL}}(q_T\|p_T) \tag{36}$$

$$= D_{\mathrm{KL}}(q_T\|p_T) + \int_0^T \frac{\partial D_{\mathrm{KL}}(q_t\|p_t)}{\partial t} dt. \tag{37}$$

The continuity equation, which is identical to the Fokker–Planck equation with zero diffusion coefficient, holds between likelihood and the vector field.

$$\frac{\partial q_t(\mathbf{x})}{\partial t} = -\mathrm{div}\left(\boldsymbol{u}(\mathbf{x})q_t(\mathbf{x})\right), \quad \frac{\partial p_t(\mathbf{x})}{\partial t} = -\mathrm{div}\left(\boldsymbol{v}_{\boldsymbol{\theta}}(\mathbf{x}, t)p_t(\mathbf{x})\right). \tag{38}$$

Then, we can rewrite the integral part of Equation (37) as

$$\frac{\partial D_{\mathrm{KL}}(q_t\|p_t)}{\partial t} \tag{39}$$

$$= \frac{\partial}{\partial t} \int q_t(\mathbf{x})[\log q_t(\mathbf{x}) - \log p_t(\mathbf{x})]d\mathbf{x} \tag{40}$$

$$= \int \frac{\partial q_t(\mathbf{x})}{\partial t} \log \frac{q_t(\mathbf{x})}{p_t(\mathbf{x})}d\mathbf{x} + \int \frac{\partial q_t(\mathbf{x})}{\partial t} d\mathbf{x} - \int \frac{q_t(\mathbf{x})}{p_t(\mathbf{x})} \frac{\partial p_t(\mathbf{x})}{\partial t} d\mathbf{x} \tag{41}$$

$$= -\int \mathrm{div}\left(\boldsymbol{u}(\mathbf{x})q_t(\mathbf{x})\right) \log \frac{q_t(\mathbf{x})}{p_t(\mathbf{x})}d\mathbf{x} + \frac{\partial}{\partial t} \int q_t(\mathbf{x})d\mathbf{x} - \int \frac{q_t(\mathbf{x})}{p_t(\mathbf{x})} \mathrm{div}\left(\boldsymbol{v}_{\boldsymbol{\theta}}(\mathbf{x}, t)p_t(\mathbf{x})\right)d\mathbf{x} \tag{42}$$

$$= \int (\boldsymbol{u}(\mathbf{x})q_t(\mathbf{x}))^T \nabla \log \frac{q_t(\mathbf{x})}{p_t(\mathbf{x})}d\mathbf{x} - \int (\boldsymbol{v}_{\boldsymbol{\theta}}(\mathbf{x}, t)p_t(\mathbf{x}))^T \nabla \frac{q_t(\mathbf{x})}{p_t(\mathbf{x})}d\mathbf{x} \tag{43}$$

$$= \int q_t(\mathbf{x})[\boldsymbol{u}(\mathbf{x}) - \boldsymbol{v}_{\boldsymbol{\theta}}(\mathbf{x}, t)]^T [\nabla \log q_t(\mathbf{x}) - \nabla \log p_t(\mathbf{x})]d\mathbf{x}, \tag{44}$$

where we use $\int q_t(\mathbf{x})d\mathbf{x} = 1$, and integral by parts with assumptions of $\lim_{\|\mathbf{x}\|_2 \to \infty} \boldsymbol{u}(\mathbf{x})q_t(\mathbf{x}) \log \frac{q_t(\mathbf{x})}{p_t(\mathbf{x})} = 0$ and $\lim_{\|\mathbf{x}\|_2 \to \infty} \boldsymbol{v}_{\boldsymbol{\theta}}(\mathbf{x}, t)p_t(\mathbf{x}) \log \frac{q_t(\mathbf{x})}{p_t(\mathbf{x})} = 0$ in Equation (42). Then, by defining $\mathcal{J}_{\mathrm{ODE}}$ as

$$\mathcal{J}_{\mathrm{ODE}} = \int_0^T \mathbb{E}_{\mathbf{x}} \left[ [\boldsymbol{u}(\mathbf{x}) - \boldsymbol{v}_{\boldsymbol{\theta}}(\mathbf{x}, t)]^T [\nabla \log q_t(\mathbf{x}) - \nabla \log p_t(\mathbf{x})] \right] dt, \tag{45}$$

we have

$$D_{\mathrm{KL}}(q_0\|p_0) = D_{\mathrm{KL}}(q_T\|p_T) + \mathcal{J}_{\mathrm{ODE}}. \tag{46}$$

Furthermore, we have the upper bound by Cauchy–Schwarz inequality as

$$\left( \int q_t(\mathbf{x})[\boldsymbol{u}(\mathbf{x}) - \boldsymbol{v_\theta}(\mathbf{x}, t)]^T [\nabla \log q_t(\mathbf{x}) - \nabla \log p_t(\mathbf{x})] d\mathbf{x} \right)^2 \tag{47}$$

$$\leq \int q_t(\mathbf{x}) \|\boldsymbol{u}(\mathbf{x}) - \boldsymbol{v_\theta}(\mathbf{x}, t)\|_2^2 d\mathbf{x} + \int q_t(\mathbf{x}) \|\nabla \log q_t(\mathbf{x}) - \nabla \log p_t(\mathbf{x})\|_2^2 d\mathbf{x}. \tag{48}$$

By defining $\mathcal{J}_{\mathrm{FM}}$ and $\mathcal{J}_{\mathrm{Fisher}}$ as

$$\mathcal{J}_{\mathrm{FM}} = \int_0^T \mathbb{E}_{\mathbf{x}} \left[ \|\boldsymbol{u}(\mathbf{x}) - \boldsymbol{v_\theta}(\mathbf{x}, t)\|_2^2 \right] dt, \tag{49}$$

$$\mathcal{J}_{\mathrm{Fisher}} = \int_0^T \mathbb{E}_{\mathbf{x}} \left[ \|\nabla \log q_t(\mathbf{x}) - \nabla \log p_t(\mathbf{x})\|_2^2 \right] dt, \tag{50}$$

we have

$$D_{\mathrm{KL}}(q_0 \| p_0) = D_{\mathrm{KL}}(q_T \| p_T) + \mathcal{J}_{\mathrm{ODE}} \leq D_{\mathrm{KL}}(q_T \| p_T) + \sqrt{\mathcal{J}_{\mathrm{FM}}} \cdot \sqrt{\mathcal{J}_{\mathrm{Fisher}}}. \tag{51}$$
$$\square$$

Next, we derive the time derivative of the score function. From Equation (38), we have

$$\frac{\partial \nabla \log q_t(\mathbf{x})}{\partial t} = \nabla \left( \frac{1}{q_t(\mathbf{x})} \frac{\partial q_t(\mathbf{x})}{\partial t} \right) \tag{52}$$

$$= \nabla \left[ \frac{1}{q_t(\mathbf{x})} \left( -q_t(\mathbf{x}) \mathrm{div}\, \boldsymbol{u}_t(\mathbf{x}) - \boldsymbol{u}_t(\mathbf{x})^T \nabla q_t(\mathbf{x}) \right) \right] \tag{53}$$

$$= -\nabla \mathrm{div}\, \boldsymbol{u}_t(\mathbf{x}) - \nabla \boldsymbol{u}_t(\mathbf{x})^T \nabla \log q_t(\mathbf{x}) - \nabla^2 \log q_t(\mathbf{x}) \boldsymbol{u}_t(\mathbf{x}), \tag{54}$$

$$\frac{\partial \nabla \log p_t(\mathbf{x})}{\partial t} = -\nabla \mathrm{div}\, \boldsymbol{v}_\theta(\mathbf{x}, t) - \nabla \boldsymbol{v}_\theta(\mathbf{x}, t)^T \nabla \log p_t(\mathbf{x}) - \nabla^2 \log p_t(\mathbf{x}) \boldsymbol{v}_\theta(\mathbf{x}, t). \tag{55}$$

Then, we can calculate the time derivative of the score function by chain rule.

$$\frac{d \nabla \log q_t(\mathbf{x})}{dt} \tag{56}$$

$$= \frac{\partial \nabla \log q_t(\mathbf{x})}{\partial \mathbf{x}} \frac{\partial \mathbf{x}}{\partial t} + \frac{\partial \nabla \log q_t(\mathbf{x})}{\partial t} \tag{57}$$

$$= \nabla^2 \log q_t(\mathbf{x}) \boldsymbol{u}_t(\mathbf{x}) - \nabla \mathrm{div}\, \boldsymbol{u}_t(\mathbf{x}) - \nabla \boldsymbol{u}_t(\mathbf{x})^T \nabla \log q_t(\mathbf{x}) - \nabla^2 \log q_t(\mathbf{x}) \boldsymbol{u}_t(\mathbf{x}) \tag{58}$$

$$= -\nabla \mathrm{div}\, \boldsymbol{u}_t(\mathbf{x}) - \nabla \boldsymbol{u}_t(\mathbf{x})^T \nabla \log q_t(\mathbf{x}), \tag{59}$$

$$\frac{d \nabla \log p_t(\mathbf{x})}{dt} \tag{60}$$

$$= \frac{\partial \nabla \log p_t(\mathbf{x})}{\partial \mathbf{x}} \frac{\partial \mathbf{x}}{\partial t} + \frac{\partial \nabla \log p_t(\mathbf{x})}{\partial t} \tag{61}$$

$$= \nabla^2 \log p_t(\mathbf{x}) \boldsymbol{u}_t(\mathbf{x}) - \nabla \mathrm{div}\, \boldsymbol{v}_\theta(\mathbf{x}, t) - \nabla \boldsymbol{v}_\theta(\mathbf{x}, t)^T \nabla \log p_t(\mathbf{x}) - \nabla^2 \log p_t(\mathbf{x}) \boldsymbol{v}_\theta(\mathbf{x}, t). \tag{62}$$

Therefore, we have

$$\frac{d(\nabla \log p_t(\mathbf{x}) - \nabla \log q_t(\mathbf{x}))}{dt} \tag{63}$$

$$= -\left( \nabla \mathrm{div}\, \boldsymbol{v}_\theta(\mathbf{x}, t) - \nabla \mathrm{div}\, \boldsymbol{u}_t(\mathbf{x}) \right) - \left( \nabla \boldsymbol{v}_\theta(\mathbf{x}, t)^T \nabla \log p_t(\mathbf{x}) - \nabla \boldsymbol{u}_t(\mathbf{x})^T \nabla \log q_t(\mathbf{x}) \right) \tag{64}$$

$$- \nabla^2 \log p_t(\mathbf{x}) \left( \boldsymbol{v}_\theta(\mathbf{x}, t) - \boldsymbol{u}_t(\mathbf{x}) \right) \tag{65}$$

Then, we prove Theorem 3.1. By integrating Equation (63), we have

$$\nabla \log p_t(\mathbf{x}) - \nabla \log q_t(\mathbf{x}) = \nabla \log p_T(\mathbf{x}) - \nabla \log q_T(\mathbf{x}) - \int_t^T \left( \nabla \mathrm{div}\, \boldsymbol{v}_\theta(\mathbf{x}, s) - \nabla \mathrm{div}\, \boldsymbol{u}_s(\mathbf{x}) \right) ds \tag{66}$$

$$- \int_t^T \left( \nabla \boldsymbol{v}_\theta(\mathbf{x}, s)^T \nabla \log p_s(\mathbf{x}) - \nabla \boldsymbol{u}_s(\mathbf{x})^T \nabla \log q_s(\mathbf{x}) \right) ds \tag{67}$$

$$- \int_t^T \nabla^2 \log p_s(\mathbf{x}) \left( \boldsymbol{v}_\theta(\mathbf{x}, s) - \boldsymbol{u}_s(\mathbf{x}) \right) ds. \tag{68}$$

Given a sample $\mathbf{x}_0$, we can rewrite the second integral term using

$$\nabla \boldsymbol{v}_\theta(\mathbf{x}, s)^T \nabla \log p_s(\mathbf{x}) - \nabla \boldsymbol{u}_s(\mathbf{x})^T \nabla \log q_s(\mathbf{x}) \tag{69}$$

$$= [\nabla \boldsymbol{v}_\theta(\mathbf{x}, s) - \nabla \boldsymbol{u}_s(\mathbf{x}|\mathbf{x}_0)]^T [\nabla \log p_s(\mathbf{x}) - \nabla \log q_s(\mathbf{x})] \tag{70}$$

$$+ \boldsymbol{u}_s(\mathbf{x}|\mathbf{x}_0)^T [\nabla \log p_s(\mathbf{x}) - \nabla \log q_s(\mathbf{x})] \tag{71}$$

$$+ [\nabla \boldsymbol{v}_\theta(\mathbf{x}, s) - \nabla \boldsymbol{u}_s(\mathbf{x})]^T \nabla \log q_s(\mathbf{x}). \tag{72}$$

Then, from triangle inequality, we have

$$\|\nabla \log p_t(\mathbf{x}) - \nabla \log q_t(\mathbf{x})\|_2 \tag{73}$$

$$\leq \|\nabla \log p_0(\mathbf{x}) - \nabla \log q_0(\mathbf{x})\|_2 + \int_t^T \|\nabla \mathrm{div}\, \boldsymbol{v}_\theta(\mathbf{x}, s) - \nabla \mathrm{div}\, \boldsymbol{u}_s(\mathbf{x})\|_2\, ds \tag{74}$$

$$+ \int_t^T \|\nabla \boldsymbol{v}_\theta(\mathbf{x}, s) - \nabla \boldsymbol{u}_s(\mathbf{x}|\mathbf{x}_0)\|_F \cdot \|\nabla \log p_s(\mathbf{x}) - \nabla \log q_s(\mathbf{x})\|_2 ds \tag{75}$$

$$+ \int_t^T \|\boldsymbol{u}_s(\mathbf{x}|\mathbf{x}_0)\|_2 \cdot \|\nabla \log p_s(\mathbf{x}) - \nabla \log q_s(\mathbf{x})\|_2 ds \tag{76}$$

$$+ \int_t^T \|\nabla \boldsymbol{v}_\theta(\mathbf{x}, s) - \nabla \boldsymbol{u}_s(\mathbf{x})\|_F \cdot \|\nabla \log q_s(\mathbf{x})\|_2 ds \tag{77}$$

$$+ \int_t^T \|\nabla^2 \log p_s(\mathbf{x})\|_F \cdot \|\boldsymbol{v}_\theta(\mathbf{x}, s) - \boldsymbol{u}_s(\mathbf{x})\|_2 ds \tag{78}$$

$$\leq \|\nabla \log p_0(\mathbf{x}) - \nabla \log q_0(\mathbf{x})\|_2 + \int_t^T \delta_3 ds \tag{79}$$

$$+ \int_t^T (\delta_2 + \|\boldsymbol{u}_s(\mathbf{x}|\mathbf{x}_0)\|_2) \cdot \|\nabla \log p_s(\mathbf{x}) - \nabla \log q_s(\mathbf{x})\|_2 ds \tag{80}$$

$$+ \int_t^T \delta_2 \|\nabla \log q_s(\mathbf{x})\|_2 ds + \int_t^T \delta_1 C ds. \tag{81}$$

By replacing each term with the following functions

$$\alpha(t) = \|\nabla \log p_0(\mathbf{x}) - \nabla \log q_0(\mathbf{x})\|_2 + \int_t^T \delta_3 + \delta_2 \|\nabla \log q_s(\mathbf{x})\|_2 + \delta_1 C ds, \tag{82}$$

$$\beta(t) = \delta_2 + \|\boldsymbol{u}_s(\mathbf{x}|\mathbf{x}_0)\|_2, \tag{83}$$

$$\gamma(t) = \|\nabla \log p_t(\mathbf{x}) - \nabla \log q_t(\mathbf{x})\|_2, \tag{84}$$

we have

$$\gamma(t) \leq \alpha(t) + \int_t^T \beta(s)\gamma(s) ds. \tag{85}$$

By Gronwall's inequality as integral form, we have an upper bound of a solution of Equation (85) as

$$\gamma(t) \leq \alpha(t) + \int_t^T \alpha(s)\beta(s) \exp\left(\int_t^s \beta(r) dr\right) ds. \tag{86}$$

Finally, by defining $U(t, \delta_1, \delta_2, \delta_3, C, q_t)$ as

$$U(t, \delta_1, \delta_2, \delta_3, C, q_t) = \mathbb{E}_{\mathbf{x}_0} \left[ \left( \alpha(t) + \int_t^T \alpha(s)\beta(s) \exp\left(\int_t^s \beta(r) dr\right) ds \right)^2 \right], \tag{87}$$

and we have the following inequality.

$$\gamma(t)^2 = \|\nabla \log p_t(\mathbf{x}) - \nabla \log q_t(\mathbf{x})\|_2 \leq U(t, \delta_1, \delta_2, \delta_3, C, q_t). \tag{88}$$

# B  IMPLEMENTATION DETAILS

## B.1  IMPLEMENTATION DETAILS OF THE EXPERIMENTS ON 2D DATASETS

We follow Tong et al. (2024) for the implementation. We use four layers MLP with $64$ hidden units and SELU activation. We use the AdamW (Loshchilov, 2017) optimizer with the learning rate of $0.001$, batch-size of $256$, training iterations of $2000$, $\sigma_{\min} = 0.01$, and $\lambda_{\mathrm{div}} = 0.001$.

### B.2 Implementation details of the experiments on image datasets

We use UNet architecture (Ronneberger et al., 2015) as the neural network $v_\theta$. Due to the upsampling and downsampling modules in UNet, we padded MNIST images by 2 zero pixels vertically and horizontally such that the size of images is to the size of $32 \times 32$.

On MNIST and CIFAR-10, we followed the setting of Tong et al. (2024). As UNet setting, we use 2 blocks with 128 channels and a dropout of 0.1. We also use AdamW optimizer with the learning rate of $2 \times 10^{-4}$, batch-size of 128, $\sigma_{\min} = 0.0$, and $\lambda_{\mathrm{div}} = 0.001$. We set the training iterations to 100k and 400k on MNIST and CIFAR-10, respectively. Additionally, we calculate the exponential moving average (EMA) with a decay of 0.9999 for the parameter of UNet, and we use the EMA parameter for the inference phase.

On ImageNet32$\times$32, we followed the original setting of Lipman et al. (2023). As UNet setting, we use 3 blocks with 128 channels and no dropout. We also use AdamW optimizer with the learning rate of $1 \times 10^{-4}$, batch-size of 1024, and 250k iterations. Other setting is the same as MNIST and CIFAR-10.

## C Qualitative results

Figures 2, 3 and (4) show the generated images by flow matching and our proposed method trained on MNIST, CIFAR-10, and ImageNet32$\times$32, respectively. Images of each row were generated with the same random seed. Although each sample pair has a few qualitative differences, we observe no qualitative differences on average.

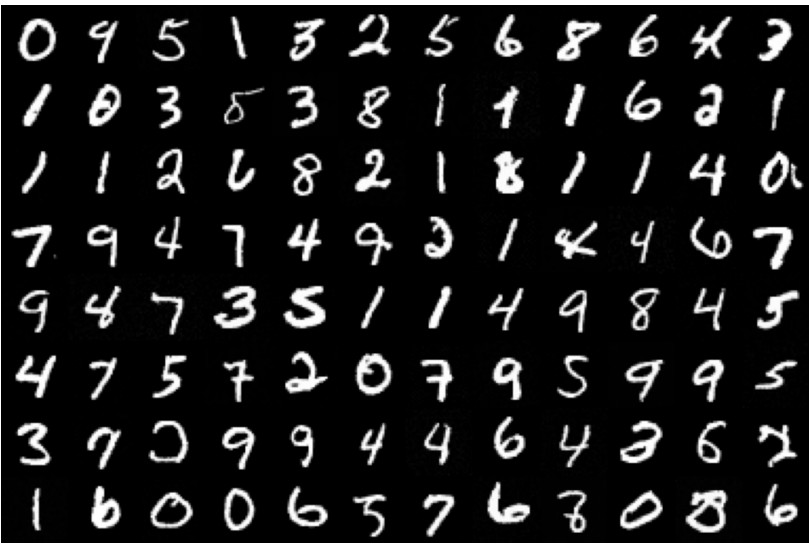

Flow matching

Ours

Figure 2: The generated images by flow matching and our proposed method trained on MNIST.

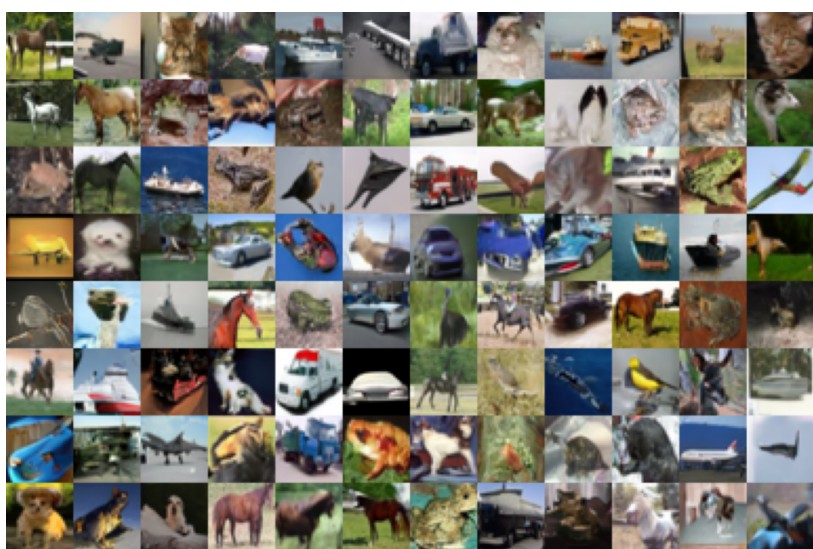

Flow matching

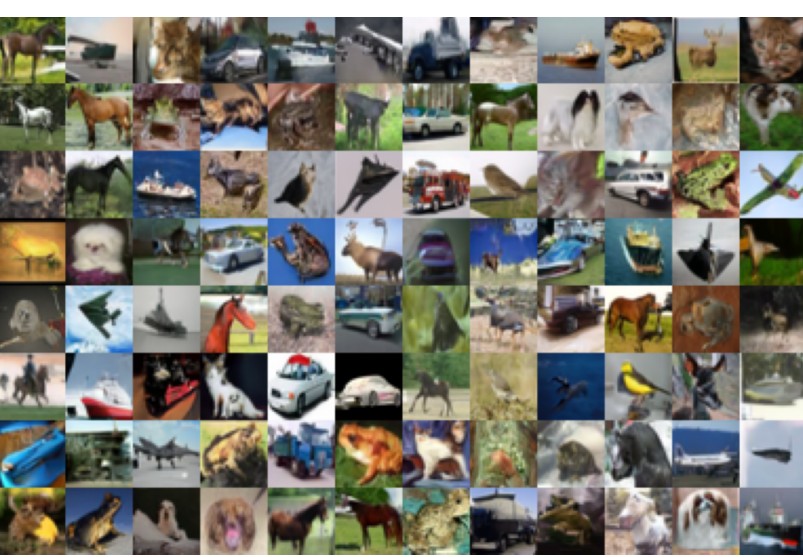

Ours

Figure 3: The generated images by flow matching and our proposed method trained on CIFAR-10.

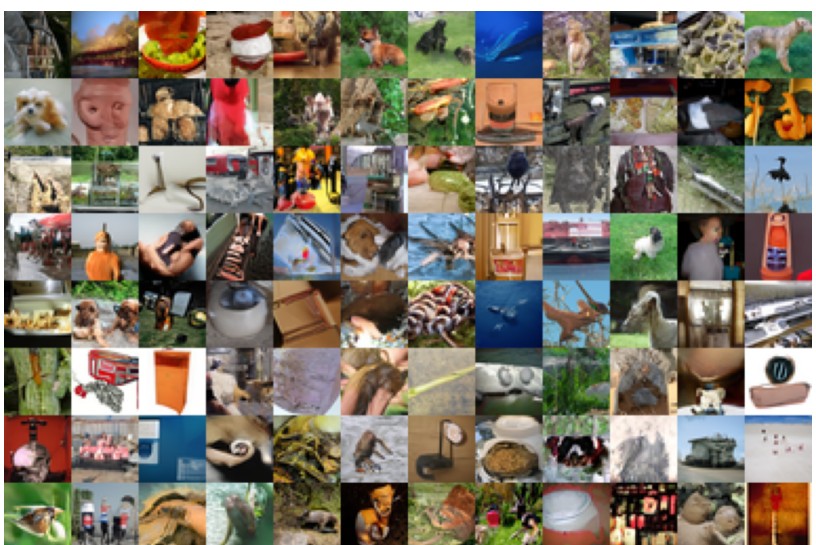

Flow matching

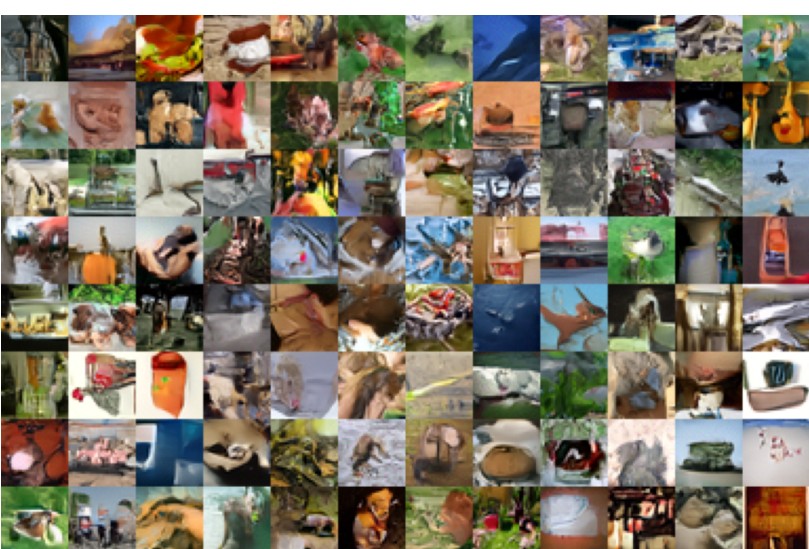

Ours

Figure 4: The generated images by flow matching and our proposed method trained on ImageNet32×32.

