# OpenReview forum: "Maximum Likelihood Estimation for Flow Matching by Direct Second-order Trace Objective"
_ICLR.cc/2025/Conference — Submitted to ICLR 2025_

### Official Review · Reviewer_YF1h · 2024-11-03

**Soundness:** 4
**Presentation:** 3
**Contribution:** 4
**Rating:** 6
**Confidence:** 4

**Summary:**

The authors propose directly minimizing a high-order objective, specifically the second-order objective, to overcome the limitations of previous methods that focused only on minimizing upper bounds. This new approach aims to optimize likelihood more effectively by reducing the KL divergence between the data distribution and the generated distribution. Experimental results show that the proposed method performs better than existing flow matching techniques on both 2D synthetic datasets and high-dimensional image datasets. So,
to summary, this paper has
1.Introduced a direct minimization technique for high-order objectives in flow matching, which enhances maximum likelihood estimation.
2.Utilized the gradient of the conditional vector field to calculate the second-order objective without needing simulations, improving computational efficiency.
3.Provided empirical evidence that the proposed method leads to better likelihood compared to previous approaches on several datasets.

**Strengths:**

1.The proposed method minimizes the KL divergence by directly addressing the second-order objective, offering a more reliable optimization of likelihood compared to earlier methods.
2.The approach demonstrates robustness across various types of datasets, including 2D synthetic and high-dimensional image datasets, indicating its scalability and versatility.

**Weaknesses:**

1.The method requires explicit computation of the second-order objective, which can be computationally intensive for very high-dimensional datasets, potentially limiting its applicability to extremely large-scale cases.
2.More details about experimental settings, such as learning rate and number of training epochs, need to be provided.
3.The tables in the paper have a lot of empty space below them, which affects the overall formatting. The layout of the paper should be reorganized for better presentation.

**Questions:**

refer to the question above

---

> ### Author Response · Authors · 2024-11-26
>
> We appreciate the reviewer for the thoughtful comments.
>
> 1. As the reviewer pointed out, our proposed method increases the computation cost to calculate the second-order gradients. The increment is based on the dimension of the model’s hidden layers rather than the data dimension, and the training time is proportional to the hidden dimension. So, we believe that applying our method to large models is not realistically impossible. We have added this discussion in Sec 6.
>
> 2. We have provided the experimental settings in the first paragraph of Sec 5.1, 5.2, and Appendix B.
>
> 3. As the reviewer pointed out, the margin below Table 5 was large. We have reduced the margins below Table 5 by some adjustments of text. For the record, we have not modified the ICLR’s style file.

---

### Official Review · Reviewer_TrHv · 2024-11-07

**Soundness:** 3
**Presentation:** 3
**Contribution:** 2
**Rating:** 3
**Confidence:** 3

**Summary:**

The paper proposes a method for directly minimizing high-order objectives in Maximum Likelihood Estimation (MLE) for flow matching models. The proposed method directly minimizes the higher-order objectives, leading to improved likelihood estimation. The effectiveness of this approach is demonstrated through experiments on 2D synthetic datasets and high-dimensional image datasets, showing superior likelihood and data generation quality compared to previous works.

**Strengths:**

1.The paper introduces a novel approach that directly minimizes high-order objectives, rather than just their upper bounds, thereby theoretically reducing KL divergence more effectively.

2.The proposed method demonstrates better performance in terms of likelihood estimation (Negative Log-Likelihood) and 2-Wasserstein distance in experiments on both 2D synthetic and high-dimensional image datasets.

3.The use of Hutchinson's trace estimation method reduces the computational cost of calculating high-order objectives, making the approach more efficient.

**Weaknesses:**

1.While the paper claims that directly minimizing high-order objectives leads to better likelihood maximization, it does not provide rigorous formal guarantees or convergence proofs that this method will always outperform minimizing upper bounds in all scenarios. The theoretical analysis is somewhat limited in terms of offering strong mathematical assurances for the proposed approach's superiority.

2.The theoretical results rely on several assumptions, such as bounding the Fisher divergence by a function of high-order objectives. These assumptions may not always hold in practical applications, especially when dealing with more complex data distributions or models.

3.The experiments are incomplete: the methods compared in the paper have not been evaluated on all datasets. The method proposed in this paper did not achieve state-of-the-art (SOTA) results on CIFAR-10 and ImageNet, and on MNIST, it was only compared with one method.

**Questions:**

See weekness.

---

> ### Author Response · Authors · 2024-11-26
>
> We appreciate the reviewer for the insightful comments.
>
> 1. As the reviewer pointed out, our theoretical guarantees are limited without more strict proofs. However, since the derivation of our proposed method includes several inequalities, differentiations, and integrations, it is hard for us to prove them. Instead, we verify the effectiveness of directly minimizing the objective in the ablation study, Sec 5.2.1.
>
> 2. We think that what the reviewer pointed out will be regarding Theorem 3.1. If so, it is what we proved in Theorem 3.1, not assumption. Additionally, the assumptions presented in our paper follow the previous work (Lu et al, 2022a).
>
> 3. According to the reviewer’s pointing out, we have conducted new experiments to recover the incompleteness of the experiments. We have conducted new experiments on ImageNet during this discussion period, however, we could not find a setting where our method outperforms other methods.

---

### Official Review · Reviewer_KR6m · 2024-11-07

**Soundness:** 2
**Presentation:** 2
**Contribution:** 1
**Rating:** 3
**Confidence:** 4

**Summary:**

This paper provides a method for directly optimizing the high-order objective for flow matching.

**Strengths:**

This paper provides a method for directly optimizing the high-order objective, extending the scope of previous work.

**Weaknesses:**

1. The theoretical results are not novel.
2. The improvement is marginal.

**Questions:**

N/A

---

> ### Author Response · Authors · 2024-11-26
>
> We appreciate the reviewer for the comments.
>
> 1. As we claim in the manuscript, especially in Sec 3.3, the novelty of our method is minimizing the second-order objective directly while the previous methods minimize the upper bound of that. Moreover, we show the effectiveness of directly minimizing the objective through the experiments on 2D datasets and image datasets.
>
> 2. In the experiments on 2D datasets, our method outperforms the original flow matching in terms of NLL on all datasets. In the experiments on image datasets, our method also outperforms the original method on MNIST and CIFAR-10. On ImageNet, we could not find a setting where our method outperforms the original method in the experiments we conducted during this discussion period.

---

### Author Response · Authors · 2024-11-26

We appreciate all reviewers for their thoughtful and insightful comments.
We have modified our manuscript in the following points:
1. We have revised some notations of time in Appendix. A. Specifically, some parts of the time notation previously used 0 for noise and 1 for data. However, to align with the conventional formulation of diffusion models, we have revised it to use T for noise and 0 for data.

2. According to reviewer YF1h's comment, we have added a discussion of increasing training time in Sec.6, which we highlighted in blue.

We hope that all of the reviewers' concerns will be adequately addressed.
I look forward to engaging in further constructive discussions with the reviewers.

---

### Meta-Review · Area_Chair_Xrjr · 2024-12-13

**Metareview:**

This paper aims to optimize the high-order objective for flow matching. The paper shows the better performance of the proposed method in terms of likelihood estimation (Negative Log-Likelihood) and 2-Wasserstein distance in experiments on both 2D synthetic and high-dimensional image datasets. But reviewers find that the theoretical analysis is somewhat limited in terms of offering strong mathematical assurances. The  assumptions are also very strong that can not be verified in real world data sets. The quality of this paper is below top conference bar.

**Additional Comments On Reviewer Discussion:**

There is no discussion.  But reviewers find that the theoretical analysis is somewhat limited in terms of offering strong mathematical assurances. The  assumptions are also very strong that can not be verified in real world data sets.

---

### Decision · Program_Chairs · 2025-01-22

Reject